# Generating Informative and Diverse Conversational Responses via Adversarial Information Maximization

**Yizhe Zhang**     **Michel Galley**     **Jianfeng Gao**
**Zhe Gan**     **Xiujun Li**     **Chris Brockett**     **Bill Dolan**
Microsoft Research, Redmond, WA, USA
`{yizzhang,mgalley,jfgao,zhgan,xiul,chrisbkt,billdol}@microsoft.com`

## Abstract

Responses generated by neural conversational models tend to lack informativeness and diversity. We present a novel adversarial learning method, called Adversarial Information Maximization (AIM) model, to address these two related but distinct problems. To foster response diversity, we leverage adversarial training that allows distributional matching of synthetic and real responses. To improve informativeness, we explicitly optimize a variational lower bound on pairwise mutual information between query and response. Empirical results from automatic and human evaluations demonstrate that our methods significantly boost informativeness and diversity.

## 1 Introduction

Neural conversational models are effective in generating coherent and relevant responses [1, 2, 3, 4, etc.]. However, the maximum-likelihood objective commonly used in these neural models fosters generation of responses that *average out* the responses in the training data, resulting in the production of safe but bland responses [5].

We argue that this problem is in fact twofold. The responses of a system may be diverse but uninformative (e.g.,"I don't know", "I haven't a clue", "I haven't the foggiest", "I couldn't tell you"), and conversely informative but not diverse (e.g., always giving the same generic responses such as "I like music", but never "I like jazz"). A major challenge, then, is to strike the right balance between informativeness and diversity. On the one hand, we seek informative responses that are relevant and fully address the input query. Mathematically, this can be measured via Mutual Information (MI) [5], by computing the reduction in uncertainty about the query given the response. On the other hand, diversity can help produce responses that are more varied and unpredictable, which contributes to making conversations seem more natural and human-like.

The MI approach of [5] conflated the problems of producing responses that are informative and diverse, and subsequent work has not attempted to address the distinction explicitly. Researchers have applied Generative Adversarial Networks (GANs) [6] to neural response generation [7, 8]. The equilibrium for the GAN objective is achieved when the synthetic data distribution matches the real data distribution. Consequently, the adversarial objective discourages generating responses that demonstrate less variation than human responses. However, while GANs help reduce the level of blandness, the technique was not developed for the purpose of explicitly improving either informativeness or diversity.

We propose a new adversarial learning method, Adversarial Information Maximization (AIM), for training end-to-end neural response generation models that produce *informative* and *diverse* conversational responses. Our approach exploits adversarial training to encourage diversity, and explicitly maximizes a Variational Information Maximization Objective (VIMO) [9, 10] to produce

informative responses. To leverage VIMO, we train a backward model that generates source from target. The backward model guides the forward model (from source to target) to generate relevant responses during training, thus providing a principled approach to mutual information maximization. This work is the first application of a variational mutual information objective in text generation.

To alleviate the instability in training GAN models, we propose an embedding-based discriminator, rather than the binary classifier used in traditional GANs. To reduce the variance of gradient estimation, we leverage a deterministic policy gradient algorithm [11] and employ the discrete approximation strategy in [12]. We also employ a dual adversarial objective inspired by [13, 14, 15], which composes both source-to-target (forward) and target-to-source (backward) objectives. We demonstrate that this forward-backward model can work synergistically with the variational information maximization loss. The effectiveness of our approach is validated empirically on two social media datasets.

## 2 Method

### 2.1 Model overview

Let $\mathcal{D} = \{(S_i, T_i)\}_{i=1}^N$ denote a set of $N$ single-turn conversations, where $S_i$ represents a query (*i.e.*, source), $T_i$ is the response to $S_i$ (*i.e.*, target). We aim to learn a generative model $p_\theta(T|S)$ that produces both *informative* and *diverse* responses for arbitrary input queries.

To achieve this, we propose the Adversarial Information Maximization (AIM), illustrated in Figure 1, where ($i$) adversarial training is employed to learn the conditional distribution $p_\theta(T|S)$, so as to improve the *diversity* of generated responses over standard maximum likelihood training, and ($ii$) variational information maximization is adopted to regularize the adversarial learning process and explicitly maximize mutual information to boost the *informativeness* of generated responses.

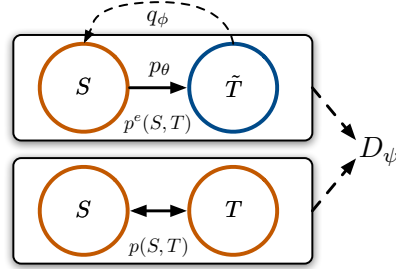

Figure 1: Overview of the Adversarial Information Maximization (AIM) model for neural response generation. Orange for real data, and blue for generated fake response. $p^e(S, T)$ represent *encoder* joint distribution, explained later.

In order to perform adversarial training, a discriminator $D_\psi(\cdot, \cdot)$ is used to distinguish real query-response pairs $(S, T)$ from generated synthetic pairs $(S, \tilde{T})$, where $\tilde{T}$ is synthesized from $p_\theta(T|S)$ given the query $S$. In order to evaluate the mutual information between $S$ and $\tilde{T}$, a *backward proposal network* $q_\phi(S|T)$ calculates a variational lower bound over the mutual information. In summary, the objective of AIM is defined as following

$$\min_\psi \max_{\theta, \phi} \mathcal{L}_{\text{AIM}}(\theta, \phi, \psi) = \mathcal{L}_{\text{GAN}}(\theta, \psi) + \lambda \cdot \mathcal{L}_{\text{MI}}(\theta, \phi), \tag{1}$$

where $\mathcal{L}_{\text{GAN}}(\theta, \psi)$ represents the objective that accounts for adversarial learning, while $\mathcal{L}_{\text{MI}}(\theta, \phi)$ denotes the regularization term corresponding to the mutual information, and $\lambda$ is a hyperparameter that balances these two parts.

### 2.2 Diversity-encouraging objective

**Generator** The conditional generator $p_\theta(T|S)$ that produces neural response $T = (y_1, \ldots, y_n)$ given the source sentence $S = (x_1, \ldots, x_m)$ and an isotropic Gaussian noise vector $Z$ is shown in Figure 2. The noise vector $Z$ is used to inject noise into the generator to prompt diversity of generated text.

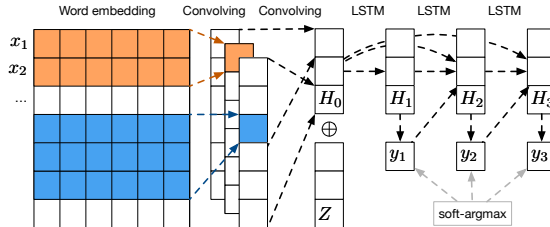

Figure 2: Illustration of the CNN-LSTM conditional generator.

Specifically, a 3-layer convolutional neural network (CNN) is employed to encode the source sentence $S$ into a fixed-length hidden vector $H_0$. A random noise vector $Z$ with the same dimension of $H_0$ is then added to $H_0$ element-wisely.

This is followed by a series of long short-term memory (LSTM) units as decoder. In our model, the $t$-th LSTM unit takes the previously generated word $y_{t-1}$, hidden state $H_{t-1}$, $H_0$ and $Z$ as input, and generates the next word $y_t$ that maximizes the probability over the vocabulary set. However, the *argmax* operation is used, instead of sampling from a multinomial distribution as in the standard LSTM. Thus, all the randomness during the generation is clamped into the noise vector $Z$, and the reparameterization trick [16] can be used (see Eqn. (4)). However, the *argmax* operation is not differentiable, thus no gradient can be backpropagated through $y_t$. Instead, we adopt the soft-argmax approximation [12] below:

$$\mathtt{onehot}(y_t) \approx \mathrm{softmax}\Big((V \cdot H_t) \cdot 1/\tau\Big), \tag{2}$$

where $V$ is a weight matrix used for computing a distribution over words. When the temperature $\tau \to 0$, the argmax operation is exactly recovered [12], however the gradient will vanish. In practice, $\tau$ should be selected to balance the approximation bias and the magnitude of gradient variance, which scales up nearly quadratically with $1/\tau$. Note that when $\tau = 1$ this recovers the setting in [8]. However, we empirically found that using a small $\tau$ would result in accumulated ambiguity when generating words in our experiment.

**Discriminator**   For the *discriminator*, we adopt a novel approach inspired by the Deep Structured Similarity Model (DSSM) [17]. As shown in Figure 3, the source sentence $S$, the synthetic response $\tilde{T}$ and the human response $T$ are all projected to an embedding space with fixed dimensionality via different CNNs[1]. The embedding network for $S$ is denoted as $W_s$, while $\tilde{T}$ and $T$ share a network $W_t$. Given these embeddings, the cosine similarities of $W_s(S)$ versus $W_t(\tilde{T})$ and $W_t(T)$ are computed, denoted as $D_\psi(T, S)$ and $D_\psi(\tilde{T}, S)$, respectively. $\psi$ represents all the parameters in the discriminator.

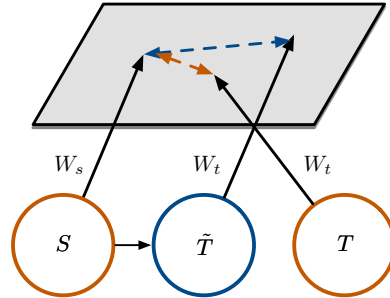

Figure 3: Embedding-based sentence discrimination.

We empirically found that separate embedding for each sentence yields better performance than concatenating $(S, T)$ pairs. Presumably, mapping $(S, T)$ pairs to the embedding space requires the embedding network to capture the cross-sentence interaction features of how relevant the response is to the source. Mapping them separately to the embedding space would divide the tasks into a sentence feature extraction sub-task and a sentence feature matching sub-task, rather than entangle them together. Thus the former might be slightly easier to train.

**Objective**   The objective of our generator is to minimize the difference between $D_\psi(T, S)$ and $D_\psi(\tilde{T}, S)$. Conversely, the discriminator tries to maximize such difference. The $\mathcal{L}_{\mathrm{GAN}}$ part in Eqn. (1) is specified as

$$\mathcal{L}_{\mathrm{GAN}}(\theta, \psi) = -\mathbb{E}_{T, \tilde{T}, S}\Big[f\Big(D_\psi(T, S) - D_\psi(\tilde{T}, S)\Big)\Big], \tag{3}$$

where $f(x) \triangleq 2\tanh^{-1}(x)$ scales the difference to deliver more smooth gradients.

Note that Eqn. (3) is conceptually related to [7] in which the discriminator loss is introduced to provide sequence-level training signals. Specifically, the discriminator is responsible for assessing both the *genuineness* of a response and the *relevance* to its corresponding source. The discriminator employed in [7] evaluates a source-target pair by operations like concatenation. However, our approach explicitly structures the discriminator to compare the embeddings using cosine similarity metrics, thus avoiding learning a neural network to match correspondence, which could be difficult. Presumably our discriminator delivers more direct updating signal by explicitly defining how the response is related to the source.

The objective in Eqn. (3) also resembles Wasserstein GAN (WGAN) [19] in that without the monotonous scaling function $f$, the discriminator $D_\psi$ can be perceived as the *critic* in WGAN with embedding-structured regularization. See details in the Supplementary Material.

To backpropagate the learning signal from the discriminator $D_\psi$ to the generator $p_\theta(T|S)$, instead of using the standard policy gradient as in [7], we consider a novel approach related to deterministic policy gradient (DPG) [11], which estimates the gradient as below:

$$\nabla_\theta \mathbb{E}_{p(\tilde{T}|S,Z)} D_\psi(\tilde{T}, S) = \mathbb{E}_{p(Z)} \nabla_{\tilde{T}} D_\psi(\tilde{T}, S) \nabla_\theta \tilde{T}(S, Z) \,, \tag{4}$$

where the expectation in Eqn. (4) approximated by Monte Carlo approximation. $\tilde{T}(S, Z)$ is the generated response, as a function of source S and randomness Z. Note that $\nabla_\theta \tilde{T}(S, Z)$ can be calculated because we use the *soft-argmax* approximation as in (2). The randomness in [7] comes from the softmax-multinomial sampling at each *local* time step; while in our approach, $\tilde{T}$ is a deterministic function of $S$ and $Z$, therefore, the randomness is *global* and separated out from the deterministic propagation, which resembles the reparameterization trick used in variational autoencoder [16]. This separation of randomness allows gradients to be deterministically backpropagated through deterministic nodes rather than stochastic nodes. Consequently, the variance of gradient estimation is largely reduced.

## 2.3 Information-promoting objective

We further seek to explicitly boost the MI between $S$ and $\tilde{T}$, with the aim of improving the *informativeness* of generated responses. Intuitively, maximizing MI allows the model to generate responses that are more specific to the source, while generic responses are largely down-weighted.

Denoting the unknown oracle joint distribution as $p(S, T)$, we aim to find an *encoder joint distribution* $p^e(S, T) = p_\theta(T|S)p(S)$ by learning a forward model $p_\theta(T|S)$, such that $p^e(S, T)$ approximates $p(S, T)$, while the mutual information under $p^e(S, T)$ remains high. See Figure 1 for illustration.

Empirical success has been achieved in [5] for mutual information maximization. However their approach is limited by the fact that the MI-prompting objective is used only during testing time, while the training procedure remains the same as the standard maximum likelihood training. Consequently, during training the model is not explicitly specified for maximizing pertinent information. The MI objective merely provides a criterion for reweighing response candidates, rather than asking the generator to produce more informative responses in the first place. Further, the hyperparameter that balances the likelihood and anti-likelihood/reverse-likelihood terms is manually selected from $(0, 1)$, which deviates from the actual MI objective, thus making the setup ad hoc.

Here, we consider explicitly maximizing mutual information $I_{p^e}(S, T) \triangleq \mathbb{E}_{p^e(S,T)} \log \frac{p^e(S,T)}{p(S)p^e(T)}$ over $p^e(S, T)$ during training. However, direct optimization of $I_{p^e}(S, T)$ is intractable. To provide a principled approach to maximizing MI, we adopt *variational information maximization* [9, 10]. The mutual information $I_{p^e}(S, T)$ under the encoder joint distribution $p^e(S, T)$ is

$$\begin{aligned}
I_{p^e}(S, T) &\triangleq \mathbb{E}_{p^e(S,T)} \log \frac{p^e(S,T)}{p(S)p^e(T)} \\
&= H(S) + \mathbb{E}_{p^e(T)} D_{KL}(p^e(S|T), q_\phi(S|T)) + \mathbb{E}_{p^e(S,T)} \log q_\phi(S|T) \\
&\geq \mathbb{E}_{p(S)} \mathbb{E}_{p_\theta(T|S)} \log q_\phi(S|T) \triangleq \mathcal{L}_{\text{MI}}(\theta, \phi) \,,
\end{aligned} \tag{5}$$

where $H(\cdot)$ denotes the entropy of a random variable, and $D_{KL}(\cdot, \cdot)$ denotes the KL divergence between two distributions. $q_\phi(S|T)$ is a *backward proposal network* that approximates the unknown $p^e(S|T)$. For this backward model $q_\phi(S|T)$, we use the same CNN-LSTM architecture as the forward model [20]. We denote the MI objective $\mathbb{E}_{p(S)} \mathbb{E}_{p_\theta(T|S)} \log q_\phi(S|T)$ as $\mathcal{L}_{\text{MI}}(\theta, \phi)$, as used in Eqn. (1).

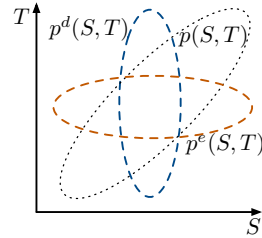

The gradient of $\mathcal{L}_{\text{MI}}(\theta, \phi)$ w.r.t. $\theta$ can be approximated by Monte Carlo samples using the REINFORCE policy gradient method [21]

$$\begin{aligned}
\nabla_\theta \mathcal{L}_{\text{MI}}(\theta, \phi) &= \mathbb{E}_{p_\theta(T|S)} \big[\log q_\phi(S|T) - b\big] \cdot \nabla_\theta \log p_\theta(T|S) \,, \\
\nabla_\phi \mathcal{L}_{\text{MI}}(\theta, \phi) &= \mathbb{E}_{p_\theta(T|S)} \nabla_\phi \log q_\phi(S|T) \,, \tag{6}
\end{aligned}$$

where $b$ is denoted as a *baseline*. Here we choose a simple empirical average for $b$ [21]. Note that more sophisticated baselines based on

Figure 4: Joint distribution matching of the query-response pairs. Details explained in Section 2.4.

neural adaptation [22] or self-critic [23] can be also employed. We complement the policy gradient objective with small proportion of likelihood-maximization loss, which was shown to stabilize the training as in [24].

As an alternative to the REINFORCE approach used in (6), we also considered using the same DPG-like approach as in (4) for approximated gradient calculation. Compared to the REINFORCE approach, the DPG-like method yields lower variance, but is less memory efficient in this case. This is because the $\mathcal{L}_{\text{MI}}(\theta, \phi)$ objective requires the gradient first back-propagated to synthetic text through all backward LSTM nodes, then from synthetic text back-propagated to all forward LSTM nodes, where both steps are densely connected. Hence, the REINFORCE approach is used in this part.

## 2.4 Dual Adversarial Learning

One issue of the above approach is that learning an appropriate $q_\phi(S|T)$ is difficult. Similar to the forward model, this backward model $q_\phi(S|T)$ may also tend to be "bland" in generating source from the target. As illustrated in Figure 4, supposing that we define a *decoder joint distribution* $p^d(S,T) = q_\phi(S|T)p(T)$, this distribution tends to be flat along $T$ axis (*i.e.*, tending to generate the same source giving different target inputs). Similarly, $p^e(S,T)$ tends to be flat along the $S$ axis as well.

To address this issue, inspired by recent work on leveraging "cycle consistency" for image generation [13, 25], we implement a dual objective that treats source and target equally, by complementing the objective in Eqn. (1) with decoder joint distribution matching, which can be written as

$$
\begin{aligned}
&\min_{\psi} \max_{\theta, \phi} \mathcal{L}_{\text{DAIM}} \\
&= -\mathbb{E}_{(T, \tilde{T}, S) \sim p_\theta^e} f(D_\psi(S, T) - D_\psi(S, \tilde{T})) \\
&\quad - \mathbb{E}_{(T, \tilde{S}, S) \sim p_\phi^d} f(D_\psi(S, T) - D_\psi(\tilde{S}, T)) \\
&\quad + \lambda \cdot \mathbb{E}_{p(S)} \mathbb{E}_{p_\theta(T|S)} \log q_\phi(S|T) \\
&\quad + \lambda \cdot \mathbb{E}_{p(T)} \mathbb{E}_{q_\phi(S|T)} \log p_\theta(T|S), \quad (7)
\end{aligned}
$$

where $\lambda$ is a hyperparameter to balance the GAN loss and the MI loss. An illustration is shown in Figure 5.

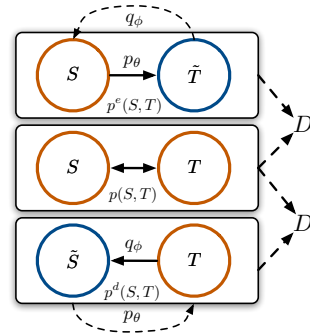

Figure 5: Dual objective for Adversarial Information Maximization (AIM).

With this dual objective, the forward and backward model are *symmetric* and *collaborative*. This is because a better estimation of the backward model $q_\phi(S|T)$ will render a more accurate evaluation of the mutual information $I_{p^e}(S, T)$, which the optimization for the forward model is based on. Correspondingly, the improvement over the forward model will also provide positive impact on the learning of the backward model. As a consequence, the forward and backward models work in a synergistic manner to simultaneously make the encoder joint distribution $p^e(S, T)$ and decoder joint distribution $p^d(S, T)$ match the oracle joint distribution $p(S, T)$. Further, as seen in Eqn. (7), the discriminators for $p^e(S, T)$ and $p^d(S, T)$ are shared. Such sharing allows the model to borrow discriminative features from both sides, and augments the synthetic data pairs (both $(S, \tilde{T})$ and $(\tilde{S}, T)$) for the discriminator. Presumably, this can facilitate discriminator training especially when source-target correspondence is difficult to learn.

We believe that this approach would also improve the generation diversity. To understand this, notice that we are maximizing a surrogate objective of $I_{p^d}(S, T)$, which can be written as

$$
I_{p^d}(S, T) = H(T) - H(T|S). \quad (8)
$$

When optimizing $\theta$, the backward model $q_\phi^d(S|T)$ is fixed and $H(T|S)$ remains constant. Thereby optimizing $I_{p^d}(S, T)$ with respect to $\theta$ can be understood as equivalently maximizing $H(T)$, which promotes the diversity of generated text.

## 3 Related Work

Our work is closely related to [5], where an *information-promoting* objective was proposed to directly optimize an MI-based objective between source and target pairs. Despite the great success of this

approach, the use of the additional hyperparameter for the anti-likelihood term renders the objective only an approximation to the actual MI. Additionally, the MI objective is employed only during testing (decoding) time, while the training procedure does not involve such an MI objective and is identical to standard maximum-likelihood training. Compared with [5], our approach considers optimizing a principled MI variational lower bound during training.

*Adversarial learning* [6, 26] has been shown to be successful in dialog generation, translation, image captioning and a series of natural language generation tasks [7, 12, 27, 28, 29, 30, 31, 32, 33, 34]. [7] leverages adversarial training and reinforcement learning to generate high quality responses. Our adversarial training differs from [7] in both the discriminator and generator design: we adopt an embedding-based structured discriminator that is inspired by the ideas from Deep Structured Similarity Models (DSSM) [17]. For the generator, instead of performing multinomial sampling at each generating step and leveraging REINFORCE-like method as in [7], we clamp all the randomness in the generation process to an initial input noise vector, and employ a discrete approximation strategy as used in [12]. As a result, the variance of gradient estimation is largely reduced.

Unilke previous work, we seek to make a conceptual distinction between informativeness and diversity, and combine the MI and GAN approaches, proposed previously, in a principled manner to explicitly render responses to be both informative (via MI) and diverse (via GAN).

Our AIM objective is further extended to a dual-learning framework. This is conceptually related to several previous GAN models in the image domain that designed for joint distribution matching [13, 25, 35, 36, 37]. Among these, our work is mostly related to the Triangle GAN [13]. However, we employ an additional VIMO as objective, which has a similar effect to that of "cycle-consistent" regularization which enables better communication between the forward and backward models. [14] also leverages a dual objective for supervised translation training and demonstrates superior performance. Our work differs from [14] in that we formulate the problem in an adversarial learning setup. It can thus be perceived as conditional distribution matching rather than seeking a regularized maximum likelihood solution.

## 4 Experiments

### 4.1 Setups

We evaluated our methods on two datasets: **Reddit** and **Twitter**. The Reddit dataset contains 2 million source-target pairs of single turn conversations extracted from Reddit discussion threads. The maximum length of sentence is 53. We randomly partition the data as (80%, 10%, 10%) to construct the training, validation and test sets. The Twitter dataset contains 7 million single turn conversations from Twitter threads. We mainly compare our results with MMI [5][2].

We evaluated our method based on *relevance* and *diversity* metrics. For relevance evaluation, we adopt BLEU [38], ROUGE [39] and three embedding-based metrics following [8, 40]. The **Greedy** metric yields the maximum cosine similarity over embeddings of two utterances [41]. Similarly, the **Average** metric [42] considers the average embedding cosine similarity. The **Extreme** metric [43] obtains sentence representation by taking the largest extreme values among the embedding vectors of all the words it contains, then calculates the cosine similarity of sentence representations.

To evaluate diversity, we follow [5] to use **Dist-1** and **Dist-2**, which is characterized by the proportion between the number of unique n-grams and total number of n-grams of tested sentence. However, this metric neglects the frequency difference of n-grams. For example, token A and token B that both occur 50 times have the same Dist-1 score (0.02) as token A occurs 1 time and token B occurs 99 times, whereas commonly the former is considered more diverse that the latter. To accommodate this, we propose to use the **Entropy** (Ent-n) metric, which reflects how evenly the empirical n-gram distribution is for a given sentence:

$$Ent = -\frac{1}{\sum_w F(w)} \sum_{w \in V} F(w) \log \frac{F(w)}{\sum_w F(w)} \,,$$

where $V$ is the set of all n-grams, $F(w)$ denotes the frequency of n-gram $w$.

Table 1: Quantitative evaluation on the Reddit dataset. (* is implemented based on [5].)

| Models | Relevance | | | | | Diversity | | |
|---|---|---|---|---|---|---|---|---|
| | BLEU | ROUGE | Greedy | Average | Extreme | Dist-1 | Dist-2 | Ent-4 |
| seq2seq | 1.85 | 0.9 | 1.845 | 0.591 | 0.342 | 0.040 | 0.153 | 6.807 |
| cGAN | 1.83 | 0.9 | 1.872 | 0.604 | 0.357 | 0.052 | 0.199 | 7.864 |
| AIM | **2.04** | **1.2** | **1.989** | **0.645** | 0.362 | 0.050 | 0.205 | 8.014 |
| DAIM | 1.93 | 1.1 | 1.945 | 0.632 | **0.366** | **0.054** | **0.220** | **8.128** |
| MMI* | 1.87 | 1.1 | 1.864 | 0.596 | 0.353 | 0.046 | 0.127 | 7.142 |
| Human | - | - | - | - | - | 0.129 | 0.616 | 9.566 |

We evaluated conditional GAN (cGAN), adversarial information maximization (AIM), dual adversarial information maximization (DAIM), together with maximum likelihood CNN-LSTM sequence-to-sequence baseline on multiple datasets. For comparison with previous state of the art methods, we also include MMI [5]. To eliminate the impact of network architecture differences, we implemented MMI-bidi [5] using our CNN-LSTM framework. The settings, other than model architectures, are identical to [5]. We performed a beam search with width of 200 and choose the hyperparameter based on performance on the validation set.

The forward and backward models were pretrained via seq2seq training. During cGAN training, we added a small portion of supervised signals to stabilize the training [24]. For embedding-based evaluation, we used a word2vec embedding trained on GoogleNews Corpus[3], recommended by [44]. For all the experiments, we employed a 3-layer convolutional encoder and an LSTM decoder as in [45]. The filter size, stride and the word embedding dimension were set to 5, 2 and 300, respectively, following [46]. The hidden unit size of $H_0$ was set to 100. We set $\lambda$ to be 0.1 and the supervised-loss balancing parameter to be 0.001. All other hyperparameters were shared among different experiments. All experiments are conducted using NVIDIA K80 GPUs.

## 4.2 Evaluation on Reddit data

**Quantitative evaluation** We first evaluated our methods on the Reddit dataset using the relevance and diversity metrics. We truncated the vocabulary to contain only the most frequent 20,000 words. For testing we used 2,000 randomly selected samples from the test set[4]. The results are summarized in Table 1. We observe that by incorporating the adversarial loss the diversity of generated responses is improved (cGAN vs. seq2seq). The relevance under most metrics (except for BLEU), increases by a small amount.

Compared MMI with cGAN, AIM and DAIM, we observe substantial improvements on diversity and relevance due to the use of the additional mutual information promoting objective in cGAN, AIM and DAIM. Table 2 presents several examples. It can be seen that AIM and DAIM produce more informative responses, due to the fact that the MI objective explicitly rewards the responses that are *predictive* to the source, and down-

| Source: | I don't suppose you have my missing socks as well? |
|---|---|
| **Human**: | You can't sleep either, I see. |
| **MMI**: | I don't have socks, but I have no idea what you're talking about. |
| seq2seq: cGAN: | I have one. I have one, but I have a pair of them. |
| **AIM**: **DAIM**: | I have one left handed. Check your pants. |

| Source: | Why does *** make such poor cell phones? Isn't that against the Japanese code? |
|---|---|
| **Human**: | They're a Korean company |
| **MMI**: | Because they use ads. |
| seq2seq: cGAN: AIM: DAIM: | I don't know how to use it. Because they are more expensive. Because they aren't in the store. Because they aren't available in Japan. |

| Source: | Why would he throw a lighter at you? |
|---|---|
| **Human**: | He was passing me it. |
| **MMI**: | Why wouldn't he? |
| seq2seq: cGAN: AIM: DAIM: | I don't know. You don't? Though he didn't use a potato. He didn't even notice that. |

Table 2: Sample outputs from different methods.

weights those that are generic and dull. Under the same hyperparameter setup, we also observe that

Table 3: Human evaluation results. Results of statistical significance are shown in bold.

| Methods | Informativeness | | | | Relevance | | | |
|---|---|---|---|---|---|---|---|---|
| | Method A | | Method B | | Method A | | Method B | |
| MMI-AIM | MMI | 0.496 | AIM | 0.504 | MMI | 0.501 | AIM | 0.499 |
| MMI-cGAN | MMI | 0.505 | cGAN | 0.495 | MMI | **0.514** | cGAN | **0.486** |
| MMI-DAIM | MMI | **0.484** | DAIM | **0.516** | MMI | 0.503 | DAIM | 0.497 |
| MMI-seq2seq | MMI | 0.510 | seq2seq | 0.490 | MMI | **0.518** | seq2seq | **0.482** |
| seq2seq-cGAN | seq2seq | 0.487 | cGAN | 0.513 | seq2seq | 0.492 | cGAN | 0.508 |
| seq2seq-AIM | seq2seq | 0.478 | AIM | 0.522 | seq2seq | 0.492 | AIM | 0.508 |
| seq2seq-DAIM | seq2seq | **0.468** | DAIM | **0.532** | seq2seq | **0.475** | DAIM | **0.525** |
| Human-DAIM | Human | **0.615** | DAIM | **0.385** | Human | **0.600** | DAIM | **0.400** |

DAIM benefits from the additional backward model and outperforms AIM in diversity, which better approximates human responses. We show the histogram of the length of generated responses in the Supplementary Material. Our models are trained until convergence. cGAN, AIM and DAIM respectively consume around 1.7, 2.5 and 3.5 times the computation time compared with our seq2seq baseline.

The distributional discrepancy between generated responses and ground-truth responses is arguably a more reasonable metric than the single response judgment. We leave it to future work.

**Human evaluation** Informativeness is not easily measurable using automatic metrics, so we performed a human evaluation on 600 random sampled sources using crowd-sourcing. Systems were paired and each pair of system outputs was randomly presented to 7 judges, who ranked them for informativeness and relevance[5]. The human preferences are shown in Table 3. A statistically significant ($p < 0.00001$) preference for DAIM over MMI is observed with respect to informativeness, while relevance judgments are on par with MMI. MMI has proved a strong baseline: the other two GAN systems are (with one exception) statistically indistinguishable from MMI, which in turn perform significantly better than seq2seq. Box charts illustrating these results can be found in the Supplementary Material.

Table 4: Quantitative evaluation on the Twitter dataset.

| Models | Relevance | | | | | Diversity | | |
|---|---|---|---|---|---|---|---|---|
| | BLEU | ROUGE | Greedy | Average | Extreme | Dist-1 | Dist-2 | Ent-4 |
| seq2seq | 0.64 | 0.62 | 1.669 | 0.54 | 0.34 | 0.020 | 0.084 | 6.427 |
| cGAN | 0.62 | 0.61 | 1.68 | 0.536 | 0.329 | 0.028 | 0.102 | 6.631 |
| AIM | **0.85** | **0.82** | **1.960** | **0.645** | **0.370** | 0.030 | 0.092 | 7.245 |
| DAIM | 0.81 | 0.77 | 1.845 | 0.588 | 0.344 | **0.032** | **0.137** | **7.907** |
| MMI | 0.80 | 0.75 | 1.876 | 0.591 | 0.348 | 0.028 | 0.105 | 7.156 |

## 4.3 Evaluation on Twitter data

We further compared our methods on the Twitter dataset. The results are shown in Table 4. We treated all dialog history before the last response in a multi-turn conversation session as a source sentence, and use the last response as the target to form our dataset. We employed CNN as our encoder because a CNN-based encoder is presumably advantageous in tracking long dialog history comparing to an LSTM encoder. We truncated the vocabulary to contain only 20k most frequent words due to limited flash memory capacity. We evaluated each methods on 2k test data.

Adversarial training encourages generating more diverse sentences, at the cost of slightly decreasing the relevance score. We hypothesize that such a decrease is partially attributable to the evaluation metrics we used. All the relevance metrics are based on *utterance-pair* discrepancy, *i.e.*, the score assesses how close the system output is to the ground-truth response. Thus, the MLE system output tends to obtain a high score despite being bland, because a MLE response by design is most "relevant"

to any random response. On the other hand, adding diversity without improving semantic relevance may occasionally hurt these relevance scores.

However the additional MI term seems to compensate for the relevance decrease and improves the response diversity, especially in Dist-$n$ and Ent-$n$ with a larger value of $n$. Sampled responses are provided in the Supplementary Material.

## 5 Conclusion

In this paper we propose a novel adversarial learning method, Adversarial Information Maximization (AIM), for training response generation models to promote informative and diverse conversations between human and dialogue agents. AIM can be viewed as a more principled version of the classical MMI method in that AIM is able to directly optimize the (lower bounder of) the MMI objective in model training while the MMI method only uses it to rerank response candidates during decoding. We then extend AIM to DAIM by incorporating a dual objective so as to simultaneously learn forward and backward models. We evaluated our methods on two real-world datasets. The results demonstrate the our methods do lead to more informative and diverse responses in comparison to existing methods.

## Acknowledgements

We thank Adji Bousso Dieng, Asli Celikyilmaz, Sungjin Lee, Chris Quirk, Chengtao Li for helpful discussions. We thank anonymous reviewers for their constructive feedbacks.

## Footnotes

[1]Note that encoders based on RNN or pure word embedding [18] are also possible, nevertheless we limit our choice to CNN in this paper.

[2]We did not compare with [8] since the code is not available, and the original training data used in [8] contains a large portion of test data, owing to data leakage.

[3]https://drive.google.com/file/d/0B7XkCwpI5KDYNlNUTTlSS21pQmM

[4]We did not use the full test set because MMI decoding is relatively slow.

[5]Relevance relates to the degree to which judges perceived the output to be semantically tied to the previous turn, and can be regarded as a constraint on informativeness. An affirmative response like "Sure" and "Yes" is relevant but not very informative.

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
