[Reviews · NeurIPS 2018]

Reviewer 1



This paper introduced a framework named adversarial information maximization (AIM) to improve the informativeness and diversity of conversational responses. The framework (a) leverages adversarial training to enable distributional matches of synthetic and real responses, thus enhancing response diversity, and (b) optimizes a variational lower bound of pairwise mutual information between queries and responses to improve response informativeness. Experimental results show the proposed framework performs well. The paper is generally clearly described. The proposed framework seeks to balance the diversity and informativeness aspects of conversational responses, which is a meaningful research problem, though I wasn't quite sure about the necessity of the dual adversarial learning objective introduced in Sec. 2.4. Below are some specific points. It'd be great if the paper could provide justification on the effectiveness of the objective described in Eq. (3), where the source (S), target (T), and synthesized target (\tilde{T}) sequences are projected to the embedding space and the objective is to minimize the difference between cosine(S,T) and cosine(S,\tilde{T}). I wonder if it works better to project (S,T) and (S,\tilde{T}) to two embeddings and then minimize the cosine similarity between the two. The physical meaning of \nabla_{\theta}\tilde{T}(S,Z) and the motivation of adding it to Eq. (4) perhaps could be more thoroughly explained. I found the backward proposal network q_{\phi}(S|T) to be interesting. Given that the response (T) can be much shorter than the query (S), I'm also curious to know about what this network is expected to learn. Sec. 2.4 seeks to address this issue with a dual objective, but I feel it perhaps could be made simpler? Why not (a) calculate how likely a synthesized response (\tilde{T}) can be paired to a query (S), and then (b) divide it by the sum of such likelihoods over all queries of the corpus (or the current mini-batch), in order to estimate q_{\phi}(S|T).

Reviewer 2



The authors present several modified GAN architectures and losses for tackling response generation in a single-turn conversational setting. In particular, the authors propose a particular architecture and loss for the discriminator, an additional mutual information component to the loss (using a variational approximation to MI and an approximate backward model), and finally a "dual" objective, which aims to treat the forward and backward models symmetrically. The authors apply their approach to two social-media style datasets, and evaluate in terms of BLEU, ROUGE, some embedding-based metrics from previous work, some diversity-targeting metrics, as well as with a human user study. Their approaches improve over baseline seq2seq, conditional GAN, and the MMI model of Li et al. (2016) in essentially all metrics. Strengths: - The authors carry out a large number of experiments with a large number of evaluation metrics (including human evaluation), and their approach generally outperforms the baselines. - The experimental comparison between cGAN, AIM, and DAIM effectively ablates at least some of the novel architectural and loss components introduced by the authors. - The authors suggest several modifications to GAN-style generation models that appear to be helpful. Weaknesses: - It would have been much better if the authors had compared their approaches with previously published results on established datasets. For instance the datasets of Li et al. (2016) could presumably have used. Even if, as in footnote 1, there are concerns about some of the other datasets, some results on these datasets could have at least been included in an appendix in order to better establish how the proposed approaches measure against previously published results. - While some of the novel architectural and loss components are effectively ablated (as above), the authors claim additional benefits of their approach that are not backed up theoretically or empirically, such as that the cosine-similarity based discriminator is superior to other discriminator architectures, and that learning with DPG is preferable to other approaches. Finally, there are some relatively small presentation-related issues: - Line 81: if Z is added to H0 element-wise this isn't a Hadamard product (which would multiply element-wise) - Line 83: it's a bit strange to claim you aren't using a standard LSTM unit just because you take the (soft) argmax of the output distribution rather than sample; as far as I can tell the LSTM architecture hasn't changed at all. - The claims throughout of proposing a "learning framework" seem overblown; this isn't a new framework, but a modified loss and architecture.

Reviewer 3



he paper proposes a combined objective of generating both informative and diverse responses in short conversations using a CNN+LSTM deep learning architecture. The objective is referred to as Adversarial Information Maximization, where informativeness is promoted by optimizing a variational lower bound on the mutual information between stimulus and response, and diversity is promoted by the adversarial goal encouraging a match between the distributions of synthesized and natural responses, under the assumption that the diversity in natural responses provides good guidance. In the descriptions of the architecture and algorithms, additional details are mentioned that are introduced to combat certain known problems, such as the instability of GAN training. Some minor improvements on some respective merit scores are observed for two datasets from Reddit and Twitter. Between the two proposed versions (basic AIM and a bi-directional option called DAIM), there are tradeoffs between informativeness and diversity. It is not clear whether such tradeoffs can be easily tuned by the parameter controlling the mixing of the two objectives. The proposed ideas are well motivated and reasonable, and apparently work to some extent in two real-world datasets. The paper is well written, and represents some incremental contribution for response generation in short conversations that are not goal-oriented. The work seems to better fit a linguistic/NLP related conference as it drills deep down to a special application scenario with some minor though well justified innovation. There is not much that can be learned and applied to problems outside the chosen use case.